# DIFFERENTIALLY PRIVATE RANDOM SPANNING TREE GENERATION

## ABSTRACT

Random spanning trees (RSTs) are a fundamental object in graph theory with wide-ranging applications in network design, reliability analysis, and randomized algorithms. However, when the underlying graph encodes sensitive information, such as private user relationships or confidential communication links, directly releasing sampled spanning trees may leak critical structural details. To address this issue, we study the problem of generating random spanning trees under differential privacy constraints. We introduce DP-RST, the first algorithmic framework for differentially private random spanning tree generation. Our method perturbs edge weights by decomposing them into binary representations and applying randomized response at the bit level, then recombining the noisy weights and sampling a spanning tree from the perturbed graph. This carefully designed pipeline injects noise while preserving the essential utility of RSTs, thereby ensuring $(\epsilon, \delta)$-DP. We further demonstrate that DP-RST achieves privacy protection with comparable computational efficiency to existing non-private RST algorithms, making it suitable for large-scale graphs. This work bridges the gap between random spanning tree generation and differential privacy, opening new directions for privacy-preserving graph algorithms.

## 1 INTRODUCTION

Random spanning trees (RSTs) are a fundamental object in graph theory with wide-ranging applications in network design, reliability analysis, and randomized algorithms (Aldous, 1990; Broder, 1989; Kelner & Madry, 2009). Given a connected graph $G = (V, E)$, a random spanning tree is a spanning tree sampled from a specific distribution over all spanning trees of $G$, often uniformly at random (Aldous, 1990) or according to edge weights (Anari et al., 2021). Efficiently generating random spanning trees has been extensively studied, with classical approaches based on random walks, determinant-based methods, and more recent fast algorithms leveraging effective resistance and combinatorial graph structures (Madry et al., 2014; Durfee et al., 2017).

Beyond classical applications (Asadpour et al., 2010; Goyal et al., 2009; Gharan et al., 2011; Fung et al., 2011), random spanning trees are increasingly used in scenarios involving sensitive user or system data, including graph machine learning (Cesa-Bianchi et al., 2010; 2013), answering connectivity queries in large-scale graphs (Chen et al., 2022b; 2025b), and fast computation of Personalized PageRank (PPR) (Jeh & Widom, 2003; Liao et al., 2022). For instance, in graph machine learning, releasing random spanning trees sampled from a social network graph could reveal whether specific friendships or interactions exist between users (Lin et al., 2022b; Epasto et al., 2022). In the case of Personalized PageRank, random spanning forests could inadvertently leak a user's closest social or professional relationships (Mazloom & Gordon, 2018; Wei et al., 2024). These scenarios highlight that protecting structural information is essential, motivating the need for differentially private random spanning trees.

Therefore, in this paper, we propose to study this fundamental and timely research question:

*Can we design privacy-preserving algorithms to generate random spanning trees while still*
*preserving the utility and efficiency of the sampled trees?*

This research question is technically novel and challenging, as it differs fundamentally from prior work on privacy-preserving algorithms for graph problems such as minimum spanning trees (Hladik

& Tetek, 2024; Pagh et al., 2025), shortest paths (Sealfon, 2016; Chen et al., 2023), and minimum cuts (Dalirrooyfard et al., 2023; Aamand et al., 2024). To the best of our knowledge, our work is the first to address the privacy-preserving computation of random spanning trees.

To address our key research question, we develop a differential privacy (DP) framework for generating random spanning trees, namely DP-RST. The construction of our differentially private weighted random spanning tree proceeds in four stages. First, each edge weight is decomposed into its binary representation so that privacy mechanisms can be applied independently at the bit level. Second, we apply randomized response to every bit, ensuring local privacy for each component. Third, the perturbed bits are recombined to reconstruct noisy edge weights while preserving overall differential privacy through composition. Finally, a random spanning tree is sampled from the perturbed weighted graph, yielding an output that balances privacy protection with structural utility.

The technical pipeline mentioned above results in a general DP framework for random spanning tree generation that achieves $(\epsilon, \delta)$-differential privacy while retaining the efficiency of standard RST algorithms. Our contributions can be summarized as follows:

- We present the first framework for generating differentially private random spanning trees, providing $(\epsilon, \delta)$-DP.

- The proposed DP-RST algorithm maintains a similar time complexity as the standard random spanning tree algorithm, making it practical for large graphs.

**Roadmap.** In Section 2, we provide a review of relevant works. In Section 3, we introduce the fundamental concepts and definitions that form the basis of our analysis. In Section 4, we present our proposed DP-RST algorithm and prove that it satisfies $(\epsilon, \delta)$-differential privacy. In Section 5, we analyze the utility of DP-RST. In Section 6, we provide the analysis of the algorithm's running time, highlighting its computational efficiency. Finally, in Section 7, we conclude our paper.

## 2 RELATED WORK

In Section 2.1, we review the related works on random spanning trees. In Section 2.2, we show the related works on differential privacy. In Section 2.3, we discuss several relevant graph problems in basic graph theory.

### 2.1 RANDOM SPANNING TREE

The random spanning tree (RST) is one of the most well-established probabilistic concepts in graph theory, with its earliest study tracing back to the 19th century (Kirchhoff, 1847). Early breakthroughs were achieved independently by Aldous (Aldous, 1990) and Broder (Broder, 1989), who proposed sampling methods based on simulating random walks over the graph. This approach yields exact samples but suffers from high time complexity, as it requires the walk to cover all edges of the graph. Subsequent research has focused on improving sampling efficiency. For instance, Kelner and Madry (Kelner & Madry, 2009) introduced a faster approach by exploiting connections between random walks and electrical network theory, and Madry et al. (Madry et al., 2014) achieved $O(m^{4/3+o(1)})$ expected time by combining random walks, effective resistance, and graph cut structure. Next, Durfee et al. (Durfee et al., 2017) further improved sampling for edge-weighted graphs using Gaussian elimination and approximate Schur complements, avoiding determinant- and random-walk-based techniques. More recent algorithms have reduced the sampling time for random spanning trees to nearly linear (Schild, 2018).

RSTs have a wide range of applications in theoretical computer science, including but not limited to approximating the traveling salesperson problem (Asadpour et al., 2010; Gharan et al., 2011) and graph sparsification (Goyal et al., 2009; Fung et al., 2011), as well as in graph data management (Chen et al., 2022b; Liao et al., 2022; Chen et al., 2025b) and machine learning (Cesa-Bianchi et al., 2010; 2013). In this work, we study the differentially private computation of random spanning trees for the first time, to the best of our knowledge.

## 2.2 Differential Privacy

Differential Privacy (DP), first formalized in (Dwork et al., 2006), has become widely recognized as the gold standard for rigorous privacy protection, with historical roots tracing back to randomized response mechanisms from the 1960s (Warner, 1965). DP provides a more reliable and provable privacy guarantee compared with conventional anonymization techniques (Sweeney, 2002; Li et al., 2006; Machanavajjhala et al., 2007), which are vulnerable to re-identification and linkage attacks that exploit external user information.

The core intuition of DP is that for any two neighboring datasets differing in only a small part (e.g., a single record or a single edge in a graph), the output of a randomized algorithm should remain statistically indistinguishable. This ensures that the presence or absence of any individual element contributes only a limited and quantifiable amount of information leakage. Such guarantees are typically achieved by injecting carefully calibrated noise into the computation, e.g., through the Gaussian (Dwork et al., 2014; Balle & Wang, 2018) or Laplace mechanisms (Dwork et al., 2014; Geng et al., 2020). Over time, several refined formulations of DP have been developed, including Rényi Differential Privacy (RDP) (Mironov, 2017; Mironov et al., 2019) and Local Differential Privacy (LDP) (Evfimievski et al., 2003; Kasiviswanathan et al., 2011), which enable tighter analysis and more flexible trade-offs between privacy and utility. These advances have enabled numerous non-trivial applications of DP across diverse fields, such as classical algorithms (Andoni et al., 2023; Li & Li, 2023; Song et al., 2023; Feng et al., 2025) and data structures (Qin et al., 2022; Ke et al., 2025), machine learning (Chaudhuri & Monteleoni, 2008; Jayaraman & Evans, 2019; Triastcyn & Faltings, 2020), large language models (Yu et al., 2022; Du et al., 2023; Mai et al., 2024), learning on graphs (Lin et al., 2022a; Olatunji et al., 2023; Sajadmanesh et al., 2023), and computer vision (Zheng et al., 2019; Zhu et al., 2020; Luo et al., 2021). Despite the rapid development and wide adoption of DP in these areas, the problem of generating random spanning trees (RSTs) under differential privacy has not been studied before. In this work, we address this novel and technically challenging application for the first time.

## 2.3 Graph Theory and Graph Problems

Graph theory is a fundamental research direction in theoretical computer science, aiming to design efficient algorithms for solving a wide range of graph problems. One of the most classical and well-studied problems is the single-source shortest path (SSSP) problem, which dates back to Dijkstra's algorithm in the 1950s (Dijksta, 1959). Since then, a large body of work (Fredman & Willard, 1990; 1993; Hagerup, 2000) has focused on improving the time complexity beyond the classical $O(m + n \log n)$ bound, where $n$ is the number of vertices and $m$ is the number of edges. Notably, Thorup developed a linear-time algorithm for undirected graphs with positive integer weights (Thorup, 1999), and subsequent work achieved $O(m + n \log \log \min\{n, C_{\max}\})$ time for directed graphs (Thorup, 2003), where $C_{\max}$ is the maximum edge weight. Several studies have also addressed graphs with negative edge weights, leading to further improvements in time complexity (Bringmann et al., 2023; Fineman, 2024). More recently, the state-of-the-art algorithm for directed graphs with nonnegative real edge weights achieves $O(m \log^{2/3} n)$ time (Duan et al., 2025), marking the first improvement over Dijkstra's algorithm in sparse graph settings. Another important problem is the maximum flow problem, which seeks the maximum amount of flow that can be sent from a source to a sink subject to edge capacity constraints and is equivalent, by the max-flow–min-cut theorem, to computing a minimum $s$–$t$ cut. Three classical algorithmic paradigms have been extensively developed: augmenting paths (Dinic, 1970; Boykov & Kolmogorov, 2004), push–relabel methods (Goldberg, 2008; Goldberg et al., 2015), and pseudoflow-based approaches (Hochbaum, 2008; Chandran & Hochbaum, 2009). A breakthrough in recent years has culminated in an almost-linear-time algorithm for maximum flow computation (Chen et al., 2022a; 2025a).

These foundational graph problems have also been studied under differential privacy. In particular, there exist differentially private algorithms for shortest paths (Sealfon, 2016; Chen et al., 2023) and for minimum cuts (Dalirrooyfard et al., 2023; Aamand et al., 2024). However, to the best of our knowledge, the problem of generating random spanning trees under differential privacy has been less explored prior to our work.

## 3 PRELIMINARY

In Section 3.1, we present the basic notations used in this paper. In Section 3.2, we show the background knowledge of the random spanning tree. In Section 3.3, we explain the basics of differential privacy.

### 3.1 NOTATIONS

Let $n, d$ be positive integers. We define $[n] := \{1, 2, \ldots, n\}$. We define $\binom{[n]}{d} := \{S \subseteq [n] : |S| = d\}$. We use $\Pr[\cdot]$ to denote the probability function. Let $T \subseteq E$ be a set of edges. We use $|T|$ to denote the cardinality of $T$.

### 3.2 RANDOM SPANNING TREE

We start by recalling the classical notion of a uniformly random spanning tree, which forms the foundation for the more advanced constructions that follow.

**Definition 3.1** (Uniformly Random Spanning Tree, (Aldous, 1990)). *Let $G = (V, E)$ be an undirected connected graph, and let a simple random walk start from an arbitrary vertex $s \in V$, continuing until every vertex has been visited. For each vertex $v \in V \setminus \{s\}$, let $e_v$ denote the edge through which the walk first entered $v$. Then $T = \{e_v \mid v \in V \setminus \{s\}\}$ forms a spanning tree of $G$, and moreover $T$ is distributed as a uniformly random spanning tree of $G$, see Algorithm 1.*

---

**Algorithm 1** Uniformly Random Spannning Tree

1: **procedure** UNIFORMLYRST, (ALDOUS, 1990)($G = (V, E)$)
2:     **for** each vertex $v \in V$ **do**
3:         $S_{v,(0)} \leftarrow \{v\}$                         ▷ Initialize singleton sets for each vertex
4:     **end for**
5:     pick an arbitrary starting vertex $u_0 \in V$
6:     $u \leftarrow u_0$
7:     $T \leftarrow \emptyset$                                 ▷ Initialize empty set of edges for the tree
8:     **while** not all vertices have been visited **do**
9:         sample the first edge $e = (u, v)$ that the random walk starting at $u$ uses to exit $S_{u,(0)}$
10:         $T \leftarrow T \cup \{e\}$                ▷ Add the edge that reaches a new vertex
11:         $u \leftarrow v$                         ▷ Move to the new vertex
12:     **end while**
13:     **return** $T$
14: **end procedure**

---

We now introduce the down-up random walk, which is a Markov chain on subsets used in more general sampling procedures.

**Definition 3.2** (Down-Up Random Walk, (Anari et al., 2019)). *Let $\mu : \binom{[n]}{k} \to \mathbb{R}_{\geq 0}$ be a distribution over $k$-subsets of $[n]$. The down-up random walk $P$ is the Markov chain on $\binom{[n]}{k}$ constructed via Algorithm 2.*

---

**Algorithm 2** Down-Up Random Walk, (Anari et al., 2019)

1: **for** $t = 0, 1, 2, \cdots$ **do**
2:     Let $T_t \in \binom{S_t}{k-1}$ be a subset of $S_t$ obtained by dropping one element of $S_t$ uniformly at random.
3:     Let $S_{t+1} = T_t \cup \{e\}$, where element $e$ is chosen with probability $\propto \mu(T_t \cup \{e\})$.
4: **end for**

---

Finally, we introduce the polynomial-generated weighted random spanning tree, which is based on the down-up random walk and generalizes uniform spanning trees to weighted distributions with provable approximation guarantees.

**Definition 3.3** (Polynomial-generated weighted random spanning tree, (Anari et al., 2021)). *Let $\mu : \binom{[n]}{k} \to \mathbb{R}_{\geq 0}$ be a density function on size $k$ subsets of $[n] = \{1, \ldots, n\}$, defining a distribution $\Pr[S] \propto \mu(S)$. Then Algorithm 3 takes a connected weighted graph $G = (V, E)$ on $n$ edges with weight function $w : E \to \mathbb{R}_{\geq 0}$ and parameter $\Delta > 0$ as input and outputs a spanning tree $T \subseteq E$ in time $O(n \log(n) \log(n/\Delta))$. The distribution of $T$ is guaranteed to be $\Delta$-close in total variation distance to the distribution $\mu$ over spanning trees of $G$ defined by*

$$\mu(T) \propto w^T.$$

*In particular, for $w(e) = 1$ for all $e \in E$, $\mu$ is the uniform distribution on spanning trees of $G$ as Definition 3.1.*

---

**Algorithm 3** Polynomial-generated weighted random spanning tree, (Anari et al., 2021)

---

1: **procedure** POLYGENRST($G = (V, E), w, \Delta$)
2:     $T \leftarrow$ UNIFORMLYRST($G$)
3:     $C \leftarrow E \setminus T$
4:     **while** $|\Pr[T] - \mu(T)| > \Delta$ **do**
5:         $e \leftarrow$ uniformly random element of $C$
6:         $T \leftarrow T \cup \{e\}$
7:         $f \in \text{cycle}(T)$ with probability $\propto 1/w_f$
8:         $T \leftarrow T \setminus \{f\}$
9:         $C \leftarrow E \setminus T$
10:     **end whilereturn** T
11: **end procedure**

---

### 3.3 DIFFERENTIAL PRIVACY

**Definition 3.4** (Differential Privacy, (Dwork et al., 2014)). *For $\epsilon > 0, \delta \geq 0$, a randomized function $\mathcal{A}$ is $(\epsilon, \delta)$-differentially private ($(\epsilon, \delta)$-DP) if for any two neighboring datasets $X \sim X'$, and any possible outcome of the algorithm $S \subset \text{Range}(\mathcal{A})$, $\Pr[\mathcal{A}(X) \in S] \leq e^{\epsilon} \Pr[\mathcal{A}(X') \in S] + \delta$.*

To better understand differential privacy in the context of graphs, we first define what it means for two graphs to be considered neighboring.

**Definition 3.5** (Edge-Neighboring Graphs). *Graphs $G = (V, E, w)$ and $G' = (V, E', w')$ are said to be edge-neighboring if they differ in the weight of exactly one edge $uv \in V^2$, with $|w_G(uv) - w_{G'}(uv)| \leq 1$, while all other edge weights remain identical.*

Next, we recall an important property of differentially private mechanisms: post-processing does not compromise privacy.

**Lemma 3.6** (Post-Processing Lemma for DP, (Dwork et al., 2014)). *Let $\mathcal{M} := \mathbb{N}^{|\chi|} \to \mathbb{R}$ be a randomized algorithm that is $(\epsilon, \delta)$-differentially private. Let $f : \mathbb{R} \to \mathbb{R}'$ be an arbitrarily random mapping. Then is $f \circ \mathcal{M} : \mathbb{N}^{|\chi|} \to \mathbb{R}'$ $(\epsilon, \delta)$-differentially private.*

We then present composition results, which describe how privacy guarantees degrade when multiple DP mechanisms are combined.

**Lemma 3.7** (Basic composition (Dwork et al., 2006)). *Given $t$ algorithms executed sequentially, where the $i$-th algorithm is $(\epsilon_i, \delta_i)$-DP for $\epsilon_i > 0$ and $\delta_i \geq 0$, the overall mechanism obtained by composing them is $(\epsilon_1 + \cdots + \epsilon_t, \delta_1 + \cdots + \delta_t)$-DP.*

For more refined guarantees when many mechanisms are composed adaptively, we refer to the advanced composition lemma.

**Lemma 3.8** (Composition lemma, (Dwork et al., 2010)). *Let $\epsilon \in (0, 1)$, and $M_1, \cdots, M_k$ be $\epsilon'$-DP, adaptively chosen mechanisms, then the composition $M_1 \circ \cdots \circ M_k$ is $(\epsilon, \delta)$-DP, where $\epsilon' = \frac{\epsilon}{\sqrt{8k \log(1/\delta)}}$.*

**Lemma 3.9** (Advanced Composition, Theorem 3.20 in page 53 of (Dwork et al., 2014)). *For all $\epsilon, \delta, \delta' \geq 0$, the class of $(\epsilon, \delta)$-DP mechanisms satisfies $(\epsilon, k\delta + \delta')$-DP under $k$-fold adaptive composition for:*

$$\epsilon' = \sqrt{2k \ln(1/\delta')}\epsilon + k\epsilon(e^\epsilon - 1)$$

Finally, we introduce the formal definition of the random response mechanism, which will be used later for bit-level perturbation of edges in the spanning tree.

**Definition 3.10** (Random response mechanism). *Let $T = (V, E)$ denote a tree with node set $V$ and edge set $E$. For each edge $e \in E$, let $g[e] \in \{0, 1\}$ denote whether edge $e$ is present ($g[e] = 1$) or absent ($g[e] = 0$) in the tree representation.*

*For any $e \in E$, let $\widetilde{g}[e]$ denote the perturbed version of $g[e]$ using the random response mechanism. Namely, for every edge $e$, we have*

$$\Pr[\widetilde{g}[e] = y] = \begin{cases} e^{\epsilon_0}/(e^{\epsilon_0} + 1), & y = g[e], \\ 1/(e^{\epsilon_0} + 1), & y = 1 - g[e]. \end{cases}$$

Let $a = e^{\epsilon_0}/(e^{\epsilon_0} + 1), b = 1/(e^{\epsilon_0} + 1)$. Since $a/b = e^{\epsilon_0}$, this implies random response can achieve $\epsilon_0$-DP.

## 4 DIFFERENTIAL PRIVATE WEIGHTED RANDOM SPANNING TREE

In Section 4.1, we introduce an important technique that decomposes the edge weights into binary representations. In Section 4.2, we show the bit-level privacy guarantee. In Section 4.3, we show how to reconstruct the perturbed edge weights. In Section 4.4, we show our algorithm to produce differentially private random spanning trees.

### 4.1 WEIGHT DECOMPOSITION

We begin by decomposing edge weights into their binary representations, which will later allow us to apply randomized response at the bit level.

**Definition 4.1** (Binary Weight Decomposition). *Let $G = (V, E, w)$ be a connected weighted graph, where $w : E \to \mathbb{Z}_{\geq 0}$ assigns nonnegative integer weights to edges. Assume that all edge weights are bounded by $k$:*

$$0 \leq w(e) \leq k, \text{for all } e \in E.$$

*For each edge $e \in E$, we represent its weight in binary form:*

$$w(e) = \sum_{i=0}^{\lceil \log_2 k \rceil - 1} b_i(e) \cdot 2^i,$$

*where $b_i(e) \in \{0, 1\}$ denotes the $i$-th bit of $w(e)$.*

*We then define a sequence of unweighted graphs:*

$$G_{(i)} = (V, E_{(i)}), \quad E_{(i)} = \{e \in E \mid b_i(e) = 1\},$$

*for each bit index $i \in \{0, \cdots, \lceil \log_2 k \rceil - 1\}$. That is, $G_{(i)}$ contains exactly those edges of $G$ whose $i$-th bit in the weight representation is $1$.*

A weighted random spanning tree in $G$ can thus be interpreted as the superposition of at most $\lceil \log_2 k \rceil$ unweighted spanning trees, each sampled from one of the $G^{(i)}$. This decomposition allows us to apply randomized response at the bit level, and later reconstruct the perturbed weighted distribution by combining the unweighted outcomes.

## 4.2 BIT-LEVEL DIFFERENTIAL PRIVACY

We then consider the privacy guarantees of a single bit in the binary decomposition of edge weights for the weighted random spanning tree.

**Lemma 4.2** (Single Bit of Edge Weight is Private, Informal Version of Lemma A.1). *If the following conditions hold:*

- *Let $\epsilon_0 \geq 0$.*

- *Let $\widetilde{b}_i(e) \in \{0, 1\}$ be the perturbed $i$-th bit of the weight of edge $e$.*

*Then, we can show that, for all edges $e \in E$ and all bit positions $i \in \{0, \ldots, \lceil \log_2 k \rceil - 1\}$, the perturbed bit $\widetilde{b}_i(e)$ is $\epsilon_0$-DP.*

## 4.3 RECONSTRUCTION OF PERTURBED WEIGHTS

Once all bits have been perturbed, the next step is to reconstruct the perturbed weights of the edges by recombining their randomized bits. Specifically, after applying bit-level randomized response on each unweighted graph $G_{(i)}$, we obtain a set of perturbed bits:

$$\widetilde{b}_i(e) \in \{0, 1\}, \quad \forall i = 0, \cdots, \ell - 1, e \in E,$$

where $\ell := \lceil \log_2 k \rceil$.

**Definition 4.3** (Reconstructed Perturbed Edge Weights). *For each edge $e \in E$, we define the reconstructed perturbed weight as*

$$\widetilde{w}(e) := \sum_{i=0}^{\ell-1} 2^i \cdot \widetilde{b}_i(e).$$

*The collection $\{\widetilde{w}(e)\}_{e \in E}$ defines a perturbed weighted graph $\widetilde{G} = (V, E, \widetilde{w})$, which aggregates the bit-level perturbations into integer valued edge weights.*

With this reconstruction in place, we can establish the overall privacy guarantee of the resulting perturbed weights.

**Lemma 4.4** (Reconstruction Preserves Differential Privacy, Informal Version of Lemma A.2). *Let $\delta \in (0, 1)$, let $\ell := \lceil \log_2 k \rceil$ be the number of bit levels per edge. If each bit-level perturbed bit $\widetilde{b}_i(e)$ is $\epsilon_0$-DP, then the reconstructed weights $\widetilde{w}(e)$ satisfy $(\epsilon, \delta)$-DP with*

$$\epsilon = \epsilon_0 \sqrt{8k \log(1/\delta)}.$$

Since $\widetilde{w}(e)$ satisfies $(\epsilon, \delta)$-DP for each edge $e$ under the single edge neighboring definition, the entire reconstructed weight $\widetilde{w}$ also satisfies $(\epsilon, \delta)$-DP.

Therefore, the reconstructed perturbed weights preserve differential privacy under composition, extending the bit-level guarantees to the full weighted graph.

## 4.4 GENERATING THE PERTURBED WEIGHTED RANDOM SPANNING TREE

After reconstructing the perturbed weights $\widetilde{w}$, we can generate a final random spanning tree using Algorithm 4.

# 5 UTILITY ANALYSIS OF DIFFERENTIAL PRIVATE WEIGHTED RANDOM SPANNING TREE

In this section, we analyze the utility of the differentially private weighted random spanning tree, quantifying how the bit-level perturbations affect the expected weights of sampled spanning trees and providing upper bounds on the deviation from the original distribution.

We begin by defining the objects and notation used throughout this section.

---

**Algorithm 4** Differentially private weighted random spanning tree

---

1: **procedure** POLYGENRST($G = (V, E), w, \Delta, \epsilon_0$)
2:     $\ell \leftarrow \lceil \log_2 k \rceil$
3:     **for** each edge $e \in E$ **do**
4:         Decompose weight: $w(e) = \sum_{i=0}^{\ell-1} b_i(e) \cdot 2^i$
5:         **for** $i = 0$ to $\ell - 1$ **do**
6:             $\widetilde{b}_i(e) = b_i(e)$, with probability $\frac{e^{\epsilon_0}}{e^{\epsilon_0}+1}$
7:             $\widetilde{b}_i(e) = 1 - b_i(e)$, with probability $\frac{1}{e^{\epsilon_0}+1}$
8:         **end for**
9:         Reconstruct perturbed weight: $\widetilde{w}(e) \leftarrow \sum_{i=0}^{\ell-1} 2^i \cdot \widetilde{b}_i(e)$
10:     **end for**
11:     $T \leftarrow$ UNIFORMLYRST($G$)
12:     $C \leftarrow E \setminus T$
13:     **while** $|\Pr[T] - \mu(T)| > \Delta$ **do**
14:         $e \leftarrow$ uniformly random element of $C$
15:         $T \leftarrow T \cup \{e\}$
16:         $f \in \text{cycle}(T)$ with probability $\propto 1/\widetilde{w}_f$
17:         $T \leftarrow T \setminus \{f\}$
18:         $C \leftarrow E \setminus T$
19:     **end whilereturn** T
20: **end procedure**

---

**Definition 5.1** (Spanning Trees and Perturbed Weights). *Let $G = (V, E)$ be a weighted graph with original edge weights $e$ for $e \in E$. Let $\widetilde{w}(e)$ denote the perturbed edge weights obtained via bit-level random response perturbation as in Algorithm 4.*

- *We use $T_* \in \mathcal{T}(G)$ to denote a spanning tree sampled w.r.t. the original weights $w$, where*

$$\Pr[T_*] \propto \prod_{e \in T_*} w(e).$$

- *We use $\widetilde{T} \in \mathcal{T}(G)$ to denote a spanning tree sampled w.r.t. the perturbed weights $\widetilde{w}$:*

$$\Pr[\widetilde{T}] \propto \prod_{e \in \widetilde{T}} \widetilde{w}(e).$$

- *For each edge $e$, let $t := \frac{e^{\epsilon_0}}{e^{\epsilon_0}+1}$ denote the probability that a single bit in the decomposition remains unchanged.*

Having set up the notation, we first analyze the accuracy of the reconstructed edge weights after perturbation.

**Lemma 5.2** (Reconstructed Weight Accuracy, Informal Version of Lemma B.1). *Let $\widetilde{w}(e) = \sum_{i=0}^{\ell-1} 2^i \cdot \widetilde{b}_i(e)$ be the reconstructed weight. Then $\Pr[\widetilde{w}(e) = w(e)] = t^\ell$, where $\ell = \lceil \log_2 k \rceil$.*

Building on this edge-level guarantee, we next extend the analysis to the entire spanning tree.

**Theorem 5.3** (Utility of DP-RST, Informal Version of Theorem B.2). *Let $T_*$ be a weighted random spanning tree sampled from true weights $w$, and $\widetilde{T}$ be sampled from perturbed weights $\widetilde{w}$. Then, for any fixed tree $T$, we have*

$$\Pr[\widetilde{T} = T_*] \geq t^{\ell \cdot |E(T_*)|} = t^{\ell \cdot (|V|-1)},$$

*where $|V| - 1$ is the number of edges in a spanning tree.*

Note that this lower bound is conservative and reflects only a worst-case guarantee. In practice, the probability that $\widetilde{T}$ coincides with $T_*$ is usually higher, since spanning trees depend on the relative ordering of edge weights rather than their exact values. As long as the order is preserved, $\widetilde{T}$ will still equal $T_*$. Thus, the bound $t^{\ell \cdot (|V|-1)}$ underestimates the actual utility of DP-RST, which is often better when perturbations are small or weight gaps are large.

# 6 RUNNING TIME ANALYSIS

In this section, we provide a detailed analysis of the running time for Algorithm 4. The running time can be devided into two main parts: bit-level randomized response on edge weights and the polynomial-generated weighted random spanning tree procedure.

## 6.1 RUNNING TIME FOR BIT-LEVEL RANDOMIZED RESPONSE

We first analyze the time complexity of bit-level randomized response on edge weights.

**Lemma 6.1** (Running Time for Bit-level Perturbation). *Let $n := |E|$ denote the number of edges in the graph, and $\ell = \lceil \log_2 k \rceil$ denote the number of bits per edge weight. Then the bit-level randomized response phase takes $O(n \cdot \ell)$ time.*

*Proof.* **Step 1: Weight decomposition.** Each edge weight $w(e)$ is decomposed into $\ell$ bits. This requires $O(\ell)$ time per edge. For all $n$ edges, this step takes $O(n \cdot \ell)$ time.

**Step 2: Bit-level randomized response.** For each bit $b_i(e)$, we apply randomized response, which is O(1) per bit. Since there are $\ell$ bits per edge and $n$ edges in total, this step also takes $O(n \cdot \ell)$ time.

**Step 3: Reconstruct perturbed weights.** Reconstruction of $\widetilde{w}(e)$ from $\ell$ bits requires $O(\ell)$ per edge, resulting in $O(n \cdot \ell)$ total time.

**Step 4: Combining.** Combining the three steps, the total running time for bit-level perturbation is
$$O(n \cdot \ell) = O(n \log k).$$
□

## 6.2 OVERALL RUNNING TIME

By combining the results of Lemma 6.1 and the known running time of the polynomial-generated weighted random spanning tree in Definition 3.3, we obtain the overall running time of Algorithm 4:
$$O(n \cdot \ell + n \log n \log(n/\Delta)) = O(n \log k + n \log n \log(n/\Delta)),$$
where $n = |E|$ is the number of edges, $\ell = \lceil \log_2 k \rceil$ is the number of bits per edge weight, and $\Delta > 0$ is the accuracy parameter of the polynomial-generated RST.

**Remark 6.2.** *We conclude this section with some observations regarding the computational aspects of DP-RST. In particular, we note the following:*

- *The bit-level randomized response phase is fully parallelizable across edges, so its runtime can be significantly reduced in practice.*

- *The polynomial-generated RST dominates the computational cost for most practical graphs, since $\log k \ll \log n \log(n/\Delta)$.*

- *The parameter $\Delta$ controls the trade-off between the accuracy of the sampled spanning tree distribution and the number of iterations in the polynomial-generated RST.*

*Overall, these observations suggest that DP-RST can achieve strong privacy guarantees while maintaining comparable computational efficiency to non-private RST algorithms, making it suitable for large-scale graphs.*

# 7 CONCLUSION

In this paper, we initiated the study of differentially private random spanning trees. To address this challenge, we proposed DP-RST, the first framework for generating random spanning trees under $(\epsilon, \delta)$-differential privacy. Our algorithm carefully introduces noise while preserving key structural properties, and it achieves comparable time complexity to standard random spanning tree generation, making it practical for large graphs. For future work, we plan to extend our results beyond uniform random spanning trees and polynomial-generated weighted random spanning trees to more general distributions. We hope this work will spark further research at the intersection of random graph algorithms and differential privacy.

## ETHICS STATEMENT

This paper does not involve human subjects, personally identifiable data, or sensitive applications. We do not foresee direct ethical risks. We follow the ICLR Code of Ethics and affirm that all aspects of this research comply with the principles of fairness, transparency, and integrity.

## REPRODUCIBILITY STATEMENT

We ensure reproducibility of our theoretical results by including all formal assumptions, definitions, and complete proofs in the appendix. The main text states each theorem clearly and refers to the detailed proofs. No external data or software is required.

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

# Appendix

**Roadmap.** In Section A, we supplement the missing proofs for Section 4. In Section B, we show the missing proofs for Section 5.

## A    MISSING PROOFS IN SECTION 4

We begin by presenting the proof for Lemma 4.2.

**Lemma A.1** (Single Bit of Edge Weight is Private, Formal Version of Lemma 4.2). *If the following conditions hold:*

- *Let $\epsilon_0 \geq 0$.*

- *Let $\widetilde{b}_i(e) \in \{0, 1\}$ be the perturbed $i$-th bit of the weight of edge $e$.*

*Then, we can show that, for all edges $e \in E$ and all bit positions $i \in \{0, \ldots, \lceil \log_2 k \rceil - 1\}$, the perturbed bit $\widetilde{b}_i(e)$ is $\epsilon_0$-DP.*

*Proof.* For each edge $e \in E$ and bit index $i \in \{0, \ldots, \lceil \log_2 k \rceil - 1\}$, $b_i(e) \in \{0, 1\}$ is the ground truth bit of the weight of edge $e$. For any neighboring graph $G'$ (differing in one edge weight), denote the corresponding bit as $b_i'(e)$. Similarly, let $\widetilde{b}_i(e)$ and $\widetilde{b}_i'(e)$ denote the perturbed bits.

We consider the following two cases to prove $\widetilde{b}_i(e)$ is $\epsilon_0$-DP.

**Case 1**. Suppose $b_i'(e) = b_i(e) = u$. Then

$$\Pr[\widetilde{b}_i(e) = u] = \frac{e^{\epsilon_0}}{e^{\epsilon_0} + 1}, \quad \Pr[\widetilde{b}_i'(e) = u] = \frac{e^{\epsilon_0}}{e^{\epsilon_0} + 1}.$$

Thus,

$$\frac{\Pr[\widetilde{b}_i(e) = u]}{\Pr[\widetilde{b}_i'(e) = u]} = 1.$$

Similarly,

$$\frac{\Pr[\widetilde{b}_i(e) = 1 - u]}{\Pr[\widetilde{b}_i'(e) = 1 - u]} = 1.$$

**Case 2**. Suppose $b_i'(e) \neq b_i(e)$. Let $b_i(e) = u$. Then

$$\frac{\Pr[\widetilde{b}_i(e) = u]}{\Pr[\widetilde{b}_i'(e) = u]} = e^{\epsilon_0}, \quad \frac{\Pr[\widetilde{b}_i(e) = 1 - u]}{\Pr[\widetilde{b}_i'(e) = 1 - u]} = e^{-\epsilon_0}.$$

Hence, for all $v \in \{0, 1\}$,

$$e^{-\epsilon_0} \leq \frac{\Pr[\widetilde{b}_i(e) = v]}{\Pr[\widetilde{b}_i'(e) = v]} \leq e^{\epsilon_0}.$$

Therefore, for every edge $e$ and bit index $i$, the perturbed bit $\widetilde{b}_i(e)$ satisfies $\epsilon_0$-differential privacy. □

Next, we show the proof for Lemma 4.4.

**Lemma A.2** (Reconstruction Preserves Differential Privacy, Formal Version of Lemma 4.4). *Let $\delta \in (0, 1)$, let $\ell := \lceil \log_2 k \rceil$ be the number of bit levels per edge. If each bit-level perturbed bit $\widetilde{b}_i(e)$ is $\epsilon_0$-DP, then the reconstructed weights $\widetilde{w}(e)$ satisfy $(\epsilon, \delta)$-DP with*

$$\epsilon = \epsilon_0 \sqrt{8k \log(1/\delta)}.$$

*Proof.* Since each $\widetilde{b}_i(e)$ is $\epsilon_0$-DP, the reconstruction corresponds to the adaptive composition of at most $\ell$ pure $\epsilon_0$-DP mechanisms. Applying the composition lemma (Lemma 3.8), we obtain that the reconstruction mechanism satisfies $(\epsilon, \delta)$-DP with $\epsilon = \epsilon_0 \sqrt{8k \log(1/\delta)}$. □

## B   MISSING PROOFS IN SECTION 5

In this section, we first prove Lemma 5.2.

**Lemma B.1** (Reconstructed Weight Accuracy, Formal Version of Lemma 5.2). *Let $\widetilde{w}(e) = \sum_{i=0}^{\ell-1} 2^i \cdot \widetilde{b}_i(e)$ be the reconstructed weight. Then*

$$\Pr[\widetilde{w}(e) = w(e)] = t^\ell,$$

*where $\ell = \lceil \log_2 k \rceil$.*

*Proof.* All bits must be correctly preserved for $\widetilde{w}(e)$ to equal $w(e)$. Since each bit is independent, then:

$$Pr[\widetilde{w}(e) = w(e)] = \prod_{i=0}^{\ell-1} \Pr[\widetilde{b}_i(e) = b_i(e)] = t^\ell.$$

$\square$

Then, we prove our main result on the utility of DP-RST, which is Theorem 5.3.

**Theorem B.2** (Utility of DP-RST, Formal Version of Theorem 5.3). *Let $T_*$ be a weighted random spanning tree sampled from true weights $w$, and $\widetilde{T}$ be sampled from perturbed weights $\widetilde{w}$. Then, for any fixed tree $T$, we have*

$$\Pr[\widetilde{T} = T_*] \geq t^{\ell \cdot |E(T_*)|} = t^{\ell \cdot (|V|-1)},$$

*where $|V| - 1$ is the number of edges in a spanning tree.*

*Proof.* For $\widetilde{T} = T_*$ to occur, all edges in $T_*$ must have their weights exactly reconstructed. By Lemma 5.2, each edge is correct with probability $t^\ell$, and edges are independent:

$$\Pr[\widetilde{T} = T_*] = \prod_{e \in T_*} \Pr[\widetilde{w}(e) = w(e)] = t^{\ell \cdot |E(T_*)|} = t^{\ell \cdot (|V|-1)}.$$

This calculation gives a lower bound on the probability of sampling $T_*$, because other configurations of perturbed weights could also lead to $\widetilde{T} = T_*$. Hence, we conclude:

$$\Pr[\widetilde{T} = T_*] \geq t^{\ell \cdot |E(T_*)|} = t^{\ell \cdot (|V|-1)}.$$

Thus, we complete the proof.                                                               $\square$

## LLM USAGE DISCLOSURE

LLMs were used only to polish language, such as grammar and wording. These models did not contribute to idea creation or writing, and the authors take full responsibility for this paper's content.

