# OpenReview forum: "Differentially Private Random Spanning Tree Generation"
_ICLR.cc/2026/Conference — ICLR 2026 Conference Withdrawn Submission_

### Official Review · Reviewer_f8KT · 2025-10-28

**Soundness:** 3
**Presentation:** 2
**Contribution:** 1
**Rating:** 2
**Confidence:** 4

**Summary:**

The paper studied differentially private generation of random spanning trees on weighted graphs. Given a weighted graph $G=(V,E,w)$, a random weighted spanning tree $T$ should be sampled with probability $\frac{w(T)}{\sum_{T\in \mathcal{T}} w(T)}$. The paper studied how to release a random ST with protections for edge-level privacy. Here, the graphs are with weights supported on $[0,1]$, and two neighboring graphs would differ by at most one edge, i.e., with weight at most $1$.

The main result of the paper is a differentially private algorithm that generates a random spanning tree with some privacy and utility guarantees. These guarantees are:
- Let $k$ be the maximum weight of the graph, the algorithm is $(\varepsilon \sqrt{k}, \delta)$-DP.
- For the utility, for any fixed random tree $T$ generated from the truly random ST algorithm, the paper showed that we have $\Pr(\tilde{T}=T)\geq (\varepsilon/\varepsilon+1)^{\ell (n-1)}$, where $\ell = \log{k}$ is the number of bits we need to represent the edge weights.

The main technique used in the paper is to write the binary expression of the edge weights and conduct the bit-wise randomized response mechanism. The privacy guarantees follow simply from advanced composition, and the utility guarantees are a simple calculation of the probability for the randomized response mechanism to match the weight of the original spanning tree.

**Strengths:**

The paper studied an interesting problem in the intersection of differential privacy and graph algorithms. The algorithm borrows some ideas from Anari et al. [STOC’21] and implements the algorithm in a DP manner. The paper also provided a good survey of results in DP and graph algorithms, and the proofs are easy to follow.

**Weaknesses:**

Despite the strengths, I have major criticisms of the paper, including the significance of the results, the techniques, and the presentation.

I find it very hard to appreciate the significance of the results. In the utility analysis, the paper basically showed that with very low probability (exponentially small), the tree generated by their algorithm will match a fixed randomly generated tree. The success probability is too small; furthermore, isn’t this also the case when privacy would fail? DP says that $Pr(ALG(G_11))<= \exp(\varepsilon)\cdot \Pr(ALG(G_2))$, which means if the adversary is quite lucky, e.g., it can see the generated tree matches the original, then it would be able to conduct a privacy attack. This analysis somehow nullified the motivation for constructing private algorithms.

The techniques are also quite straightforward: it’s basically a combination of binary expansion and randomized responses. The paper provides no insight into the algorithms they used as blackboxes (the analysis of those algorithms is not simple). Therefore, I don’t think the paper has significant technical contributions, either.

I think for a good result in DP RST, we would want some sort of distribution difference as the utility metric, e.g., the output distribution has a low TVD or KL divergence with the original. Something like this would require a much more careful analysis, though.

The issue about motivation: the paper discussed the importance of graph algorithms and differential privacy, and I agree that both are important in algorithm design and machine learning. However, I do not think there is a discussion about why we need to protect *edge privacy* for random spanning trees. Do you have an application in mind in this regard?

Finally, regarding the presentation, I think the paper needs to clarify the definitions for random spanning trees, and it should give some intuitions about why the algorithms in, e.g., Anari et al. [STOC’21], work for weighted graphs. In Section 3.2, you defined random spanning trees using an algorithm. However, I believe the most natural definition of a random spanning tree is simply to sample a spanning tree uniformly at random over all spanning trees. The algorithm, as in Aldous [SIDMA’90], provided an efficient algorithm: the tree generated from the random walk satisfies the distribution of a random spanning tree. You might want to reconsider the way you introduce random spanning trees.

Similarly, the notion of random ST for weighted graphs was never introduced, and Definition 3.3 directly talked about the poly-time algorithm for such applications.

In light of the above criticism, I regret to say that I won’t be supportive of this paper.

**Questions:**

Do you have responses to my comments in the weakness section?

---

> ### Author Response · Authors · 2025-11-23
>
> Thank you for your detailed comments. They provide clear guidance and will strengthen our work. We will include these changes in the next version.

---

### Official Review · Reviewer_izz5 · 2025-10-29

**Soundness:** 3
**Presentation:** 2
**Contribution:** 2
**Rating:** 2
**Confidence:** 3

**Summary:**

This paper studies the problem of generating random spanning trees (RSTs) from a graph under edge-weight neighboring differential privacy (DP).

To address this, the paper introduces **DP-RST**, which the authors present as the first algorithmic framework for $(\epsilon, \delta)$-differentially private random spanning tree generation. The proposed method works as follows:
1.  It assumes the input is a graph with integer edge weights bounded by a value $k$.
2.  It decomposes each edge's weight into its binary representation.
3.  It applies the randomized response mechanism to each bit of each edge weight independently to ensure local privacy at the bit level.
4.  It reconstructs "noisy" integer edge weights from these perturbed bits.
5.  Finally, it runs a standard, non-private polynomial-generated weighted RST sampling algorithm on the graph using the new, noisy weights. The privacy of the final output is guaranteed by the post-processing property of differential privacy.

The authors provide a privacy analysis proving that this method satisfies $(\epsilon, \delta)$-DP and a utility analysis that lower-bounds the probability of sampling the correct tree. They also analyze the computational complexity, showing it is comparable to non-private RST algorithms.

**Strengths:**

The primary strength of this paper is its novel problem formulation. To my knowledge, this is the first work to formally address the generation of *random spanning trees* under differential privacy. Prior work on DP graph algorithms has largely focused on deterministic optimization problems like minimum spanning trees, shortest paths, and minimum cuts. Tackling a *sampling* problem is a new and valuable direction.

The paper is clear and well-written. The proposed algorithm, DP-RST, is simple, intuitive, and clearly explained.

**Weaknesses:**

Despite the novel problem, the paper suffers from an uninformative utility analysis and a lack of experimental evidence as an alternative.
- The utility analysis focuses on a single metric: a lower bound on the probability that the perturbed algorithm samples the *exact same tree* $T_*$ that would have been sampled from the true weights. This bound is $Pr[\tilde{T}=T_{*}] \ge t^{l\cdot(|V|-1)}$, where $t < 1$. This metric is not meaningful for a *randomized* algorithm. The bound is exponentially small in the size of the graph ($|V|$) and the number of bits ($l$), suggesting the utility is effectively zero for any non-trivial graph. For context, it's not clear to me if a uniformly random spanning tree is much worse than the provided guarantees.
- There is no empirical evidence for either the computational efficiency (despite the claim of practicality) or the utility of the proposed algorithm. This is especially worrisome in light of the uninformative utility analysis.

**Questions:**

- Have the authors considered releasing noisy edge weights using a simple Laplace/Gaussian mechanism and running a non-private algorithm? There should be no dependence on the maximum edge weight in this case.
- Along the same line, is there any hope of handling fractional edge weights?
- Is it possible to prove some distributional distance between the output distribution of the non-private sampler compared to the proposed algorithm?
- A small clash of notation in the privacy analysis: I believe $\ell$ should be used as the number of bits, as $k$ is the maximum edge weight.

---

> ### Author Response · Authors · 2025-11-23
>
> We appreciate your constructive suggestions and careful review. Thank you for helping us improve our manuscript. We will incorporate these points in the next version.

---

### Official Review · Reviewer_VUSL · 2025-10-30

**Soundness:** 1
**Presentation:** 2
**Contribution:** 1
**Rating:** 2
**Confidence:** 5

**Summary:**

The paper considers the problem of generating a random spanning tree of a positivity weighted graph under edge differential privacy. They don't discuss this explicitly, but I think the goal is to sample a fixed tree with probability proportional to the product of its edge weights. Absent privacy, generating such a tree with respect to this probability distribution can be done by performing a weighted random walk on the graph. The paper writes the weights of the tree in binary, and uses randomized response on each bit to privatize the edge weights, then use the classic non-private algorithms to generate the tree. This approach is clearly private, but the algorithm seems to have no meaningful utility guarantees.

**Strengths:**

It is a nice question whether one can sample weighted random spanning trees in a DP fashion.

**Weaknesses:**

There is a big issue with the paper, namely that the algorithm doesn't provide any meaningful guarantees. Additionally, the analysis that the authors provide is flawed.

First of all, intuitively flipping each bits of the weights is a quite unnatural approach, as the weights can undergo enormous changes, for example if the leading bit is flipped. A priori, it is quite unclear that the distribution when sampling a tree from the modified graph would have anything to do with the distribution when sampling from the initial graph.

Indeed, the main utility analysis of the paper (Theorem 5.3) relies on not a single weighted edge of a given spanning tree being modified under the noise addition(!) which happens only with *exponentially* small probability $\exp(-\Omega(n\ell))$, where $\ell$ is the number of bits of the weights. A utility analysis that only works if no noise is added to the sensitive information, clearly does raise some concerns.



In fact, while not so meaningful anyway, the proof of Theorem 5.3 is incorrect. Indeed, in line 840 the paper states that the probability of sampling a given tree $T_*$ is $\prod_{e\in T_*}[w(e)=\tilde w(e)]$ where $\tilde w$ is the noised weight function. $\bigwedge_{e\in T_*}[w(e)=\tilde w(e)]$ does not imply that the sampled tree is $T_*$, and the statement seems quite likely to be false.


Some more minor concerns are that the paper does not define the problem (e.g. what is the distribution we want to sample from?), or the privacy model (what's the neighborhood relation on graphs?).

\paragraph{Questions}
Have you considered adding Laplace noise to the edges? This maybe seems like a more natural approach where you can actually prove some utility guarantees.

**Questions:**

Have you considered adding Laplace noise to the edges? This maybe seems like a more natural approach where you can actually prove some utility guarantees.

---

> ### Author Response · Authors · 2025-11-23
>
> We sincerely thank you for your time and effort in reviewing our work. Your insights are valuable to us. We will carefully address them in the next version.

---

### Official Review · Reviewer_gHxs · 2025-11-08

**Soundness:** 2
**Presentation:** 2
**Contribution:** 1
**Rating:** 0
**Confidence:** 3

**Summary:**

The paper gives algorithms for generating random spanning trees of a graph that are differentially private with respect to edge weights. Here, the graph is known and public. A change of one person's data will change the weight of one edge by at most 1.

The paper's goal is to sample random spanning trees with probability proportional to the sum of the weights of edges in T. The paper claims that such generation has wide applications in ML. The algorithm works by perturbing edge weights and then generating a random tree with respect to the perturbed edge weights. The utility statement is that with some nonzero probability, the perturbed edge weights will equal the real ones and, in that case, the algorithm will sample correctly.

**Strengths:**

The paper addresses a well-defined theoretical problem.

**Weaknesses:**

**Weaknesses**
* The utility statement was hard to parse and seems very weak
* The algorithm seems easy to simplify and improve.
* The motivation for the task is not well explained

**Detailed critiques:**

* The utility statement for the authors' algorithm, Theorem 5.3, is very weak. In fact, it is not even directly tied to the distribution on spanning trees that their algorithm generates. It simply states the (exponentially small in $|V|$) probability with which the perturbed weights for a given tree will equal the true weights. There is no discussion of what happens with the rest of the edges in the graph, nor what the bound ultimately implies for the distribution one actually generates.

* The proposed algorithm is to separately perturb each bit of the binary decomposition of each edge's weight and then interpret the lists of perturbed bits as integers. This seems wildly suboptimal—one could simply add Laplace (or similar) noise to each edge weight and get way less noise.

* Motivation: The paper lists applications of random spanning trees, but does not explain the connection between plausible sensitive data sets, these applications, and the specific privacy notion considered here. What is a plausible setting in which one would want to generate weighted RSTs based on sensitive data *and* edge-weight privacy with a known graph is the right model of each individual's contribution? (In particular, it would be good to explain why the proposed notion of privacy addresses the privacy failures cited as examples on page 1.)

**Minor comments**

* The name "edge privacy" for this particular notion is misleading—edge privacy normally refers to a definition that hides the presence or absence of a given edge, not a (small) change in its weight.

* The exact notion of privacy should be defined much sooner in the paper. In particular, saying that the algorithm "achieves $(\epsilon, \delta)$-DP" is not meaningful on its own; the adjacency notion really matters.

* Theorem 5.3 (and its copy in the appendix) is weirdly stated, since it says "for any fixed T..." but then $T_*$ and $\tilde T$ appear, which are random variables that take values in the space of (all) spanning trees for the underlying graph $G$. In its current form, it does not scan as a well-defined mathematical statement.

**Questions:**

It would help if the authors could restate their main theorem more correctly, and relate it to the task at hand.

---

> ### Author Response · Authors · 2025-11-23
>
> Thank you for your thoughtful feedback. Your comments are very helpful and highly appreciated. We will address these in the next version.

---

### Note · Authors · 2025-11-23

**Comment:**

We would like to sincerely thank all the reviewers for providing insightful comments to improve our work. After careful consideration, we decide to withdraw this paper.

**Withdrawal Confirmation:**

I have read and agree with the venue's withdrawal policy on behalf of myself and my co-authors.